# Functional Regularisation for Continual Learning with Gaussian Processes

**Michalis K. Titsias**[*]
DeepMind
mtitsias@google.com

**Jonathan Schwarz**[*]
DeepMind &
University College London
schwarzjn@google.com

**Alexander G. de G. Matthews**
DeepMind
alexmatthews@google.com

**Razvan Pascanu**
DeepMind
razp@google.com

**Yee Whye Teh**
DeepMind
ywteh@google.com

## Abstract

We introduce a framework for Continual Learning (CL) based on Bayesian inference over the function space rather than the parameters of a deep neural network. This method, referred to as functional regularisation for Continual Learning, avoids forgetting a previous task by constructing and memorising an approximate posterior belief over the underlying task-specific function. To achieve this we rely on a Gaussian process obtained by treating the weights of the last layer of a neural network as random and Gaussian distributed. Then, the training algorithm sequentially encounters tasks and constructs posterior beliefs over the task-specific functions by using *inducing point sparse Gaussian process* methods. At each step a new task is first learnt and then a summary is constructed consisting of (i) inducing inputs – a fixed-size subset of the task inputs selected such that it optimally represents the task – and (ii) a posterior distribution over the function values at these inputs. This summary then regularises learning of future tasks, through Kullback-Leibler regularisation terms. Our method thus unites approaches focused on (pseudo-)rehearsal with those derived from a sequential Bayesian inference perspective in a principled way, leading to strong results on accepted benchmarks.

## 1 Introduction

Recent years have seen a resurgence of interest in continual learning, which refers to systems that learn in an online fashion from data associated with possibly an ever-increasing number of tasks (Ring, 1994; Robins, 1995; Schmidhuber, 2013; Goodfellow et al., 2013). A continual learning system must adapt to perform well on all earlier tasks without requiring extensive re-training on previous data. There are two main challenges for continual learning (i) avoiding catastrophic forgetting, i.e. remembering how to solve earlier tasks, and (ii) scalability over the number of tasks. Other possible desiderata may include forward and backward transfer, i.e. learning new tasks faster and retrospectively improving on previously tasks.

Similarly to many recent works on continual learning (Kirkpatrick et al., 2017; Nguyen et al., 2017; Rusu et al., 2016; Li & Hoiem, 2017; Farquhar & Gal, 2018), we focus on the scenario where a sequence of supervised learning tasks are presented to a continual learning system based on a deep neural network. While most methods assume known task boundaries, our approach will be also extended to deal with unknown task boundaries. Among the different techniques proposed to address this problem, we have methods which constrain or regularise the parameters of the network to not deviate significantly from those learnt on previous tasks. This includes methods that frame continual learning as sequential approximate Bayesian inference, including EWC (Kirkpatrick et al., 2017) and VCL (Nguyen et al., 2017). Such approaches suffer from brittleness due to representation drift. That is, as parameters adapt to new tasks the values that other parameters are constrained/regularised towards become obsolete (see Section 2.5 for further discussion on this). On the other hand, we have rehearsal/replay buffer methods, which use a memory store of past observations to remember previous

---

[*]Equal contribution

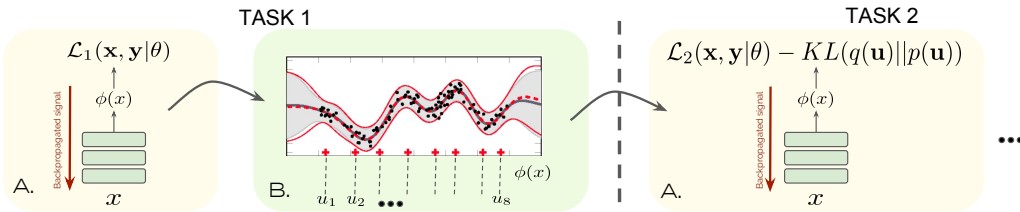

**Figure 1:** Depiction of the proposed approach. See also the provided pseudocode. When learning task 1, first, parameters of the network $\theta$ and output layer $w$ are fitted (Panel A). Afterwards, the learned GP is sparsified and inducing points $u_1, ..$ are found (Panel B). When moving to the next task the same steps are repeated. The only difference is that now the previously found summaries (in this case points $u_1, .., u_8$) are used to regularise the function (via KL-divergence term), such that the first task is not forgotten.

tasks (Robins, 1995; Robins & McCallum, 1998; Lopez-Paz et al., 2017; Rebuffi et al., 2017). While these methods tend to not suffer from brittleness, uncertainty about the unknown functions is not expressed. Furthermore, they rely on various heuristics to decide which data to store (Rolnick et al., 2018), often requiring large quantities of stored observations to achieving good performance. In this paper we will address the open problem of deriving an optimisation objective to select the best observations for storage.

In this paper, we develop a new approach to continual learning which addresses the shortcomings of both categories. It is based on approximate Bayesian inference, but on the space of functions instead of neural network parameters, so does not suffer from the aforementioned brittleness. Intuitively, while previous approaches constrain the parameters of a neural network to limit deviations from previously learnt parameters, our approach instead constrains the neural network predictions from deviating too far from those that solve previous tasks.

Effectively, our approach avoids forgetting an earlier task by memorising an approximate posterior belief over the underlying task-specific function. To implement this, we consider Gaussian processes (GPs) (Rasmussen & Williams, 2005), and make use of inducing point sparse GP methods (Csato & Opper, 2002; Titsias, 2009; Hensman et al., 2013; Bui et al., 2017b), which summarise posterior distributions over functions using small numbers of so-called inducing points. These inducing points are selected from the training set by optimising a variational objective, providing a principled mechanism to compress the dataset to a meaningful subset of fixed size. They are kept around when moving to the next task and, together with their posterior distributions, are used to regularise the continual learning of future tasks, through Kullback-Leibler regularisation terms within a variational inference framework, thus avoiding catastrophic forgetting of earlier tasks. Our approach bears similarities to replay-based approaches, with inducing points playing the role of the rehersal/replay buffer, but has two important advantages. First the approximate posterior distributions at the inducing points captures the uncertainty of the unknown function as well, rather than providing merely target values. Second, inducing points can be optimised using specialised criteria from the GP literature, achieving better performance than a random selection of observations. An intuitive depiction of our approach is given in Figure 1.

To enable our functional regularisation approach to deal with high-dimensional and complex datasets, we use a linear kernel with features parameterised by neural networks (Wilson et al., 2016). Such GPs can be understood as Bayesian neural networks, where only the weights of the last layer are treated in a Bayesian fashion, while those in earlier layers are optimised. This view allows for a more computationally efficient and accurate training procedure to be carried out in weight space, before the approximation is translated into function space where the inducing points are constructed and then used for regularising learning of future tasks. Finally, note that inducing points are also used to regularize the deep network, even though they were selected to best represent functions given by the GP. See Section 2.3 for further details.

## 2 FUNCTIONAL REGULARISATION FOR CONTINUAL LEARNING

We consider supervised learning of multiple tasks, with known task boundaries, that are processed sequentially one at a time. At each step we receive a set of examples $(X_i, \mathbf{y}_i)$ where $X_i = \{x_{i,j}\}_{j=1}^{N_i}$

are input vectors and $\mathbf{y}_i = \{y_{i,j}\}_{j=1}^{N_i}$ are output targets so that each $y_{i,j}$ is assigned to the input $x_{i,j} \in \mathbb{R}^D$. We assume the most extreme case (and challenging in terms of avoiding forgetting) where each dataset $(X_i, \mathbf{y}_i)$ introduces a new task, while less extreme cases can be treated similarly. We wish to sequentially train a shared model or representation from all tasks so that catastrophic forgetting is avoided, i.e. when the model is trained on the $i$-th task it should still provide accurate predictions for all tasks $j < i$ seen in the past. As a model we consider a deep neural network with its final hidden layer providing the feature vector $\phi(x; \theta) \in \mathbb{R}^K$ where $x$ is the input vector and $\theta$ are the model parameters. This representation is shared across tasks and $\theta$ is a task-shared parameter. To solve a specific task $i$ we additionally construct an output layer

$$f_i(x; w_i) \equiv f_i(x; w_i, \theta) = w_i^\top \phi(x; \theta), \tag{1}$$

where for simplicity we assume that $f_i(x; w_i)$ is a scalar function and $w_i$ is the vector of task-specific weights. Dealing with vector-valued functions is straightforward and is discussed in the Appendix. By placing a Gaussian prior on the output weights, $w_i \sim \mathcal{N}(w_i | 0, \sigma_w^2 I)$, we obtain a distribution over functions. While each task has its own independent/private weight vector $w_i$ the whole distribution refers to the full infinite set of tasks that can be tackled by the same feature vector $\phi(x; \theta)$. We can marginalise out $w_i$ and obtain the equivalent function space view of the model, where each task-specific function is an independent draw from a GP (Rasmussen & Williams, 2005), i.e.

$$f_i(x) \sim \mathcal{GP}(0, k(x, x')), \ k(x, x') = \sigma_w^2 \phi(x; \theta)^\top \phi(x'; \theta),$$

where the kernel function is defined by the dot product of the neural network feature vector. By assuming for now that all possible tasks are simultaneously present similarly to multi-task GPs (Bonilla et al., 2008; Álvarez et al., 2012), the joint distribution over function values and output data for all tasks is written as $\prod_i p(\mathbf{y}_i | \mathbf{f}_i) p(\mathbf{f}_i) = \prod_i p(\mathbf{y}_i | \mathbf{f}_i) \mathcal{N}(\mathbf{f}_i | \mathbf{0}, K_{X_i})$, where the vector $\mathbf{f}_i$ stores all function values for the input dataset $X_i$, i.e. $f_{i,j} = f(x_{i,j}), j = 1, \ldots, N_i$. Also the kernel matrix $K_{X_i}$ is obtained by evaluating the kernel function on $X_i$, i.e. each element $[K_{X_i}]_{j,k} = \sigma_w^2 \phi(x_{i,j}; \theta)^\top \phi(x_{i,k}; \theta)$ where $x_{i,j}, x_{i,k} \in X_i$. The form of each likelihood function $p(\mathbf{y}_i | \mathbf{f}_i)$ depends on the task, for example if the $i$-th task involves binary classification then $p(\mathbf{y}_i | \mathbf{f}_i) = \prod_{j=1}^{N_i} p(y_{i,j} | f_{i,j}) = \prod_{j=1}^{N_i} \frac{1}{1 + e^{-y_{i,j} f_{i,j}}}$ where $y_{i,j} \in \{-1, 1\}$ indicates the binary class label.

Inference in this model requires estimating each posterior distribution $p(\mathbf{f}_i | \mathbf{y}_i, X_i)$, which can be approximated by a multivariate Gaussian $\mathcal{N}(\mathbf{f}_i | \mu_i, \Sigma_i)$. Given this Gaussian we can express our posterior belief over any function value $f_{i,*}$ at some test input $x_{i,*}$ using the posterior GP (Rasmussen & Williams, 2005),

$$p(f_{i,*} | X_i, y_i) = \int p_\theta(f_{i,*} | \mathbf{f}_i) \mathcal{N}(\mathbf{f}_i | \mu_i, \Sigma_i) d\mathbf{f}_i.$$

Given that the tasks arrive one at a time, the above suggests that one way to avoid forgetting the $i$-th task is to memorise the corresponding posterior belief $\mathcal{N}(\mathbf{f}_i | \mu_i, \Sigma_i)$. While this can regularise continual learning of subsequent tasks (similarly to the more general variational framework in the next section), it can be prohibitively expensive since the non-parametric nature of the model means that for each $\mathcal{N}(\mathbf{f}_i | \mu_i, \Sigma_i)$ we need to store $O(N_i^2)$ parameters and additionally we need to keep in memory the full set of input vectors $X_i$.

Therefore, in order to reduce the time and memory requirements we would like to apply data distillation and approximate each full posterior by applying sparse GP methods. As shown next, by applying variational sparse GP inference (Titsias, 2009) in a sequential fashion we obtain a new algorithm for function space regularisation in continual learning.

## 2.1 Learning the first task

Suppose we encounter the first task with data $(X_1, \mathbf{y}_1)$. We introduce a small set $Z_1 = \{z_{1,j}\}_{j=1}^{M_1}$ of inducing inputs where each $z_{1,j}$ lives in the same space as each training input $x_{1,j}$. The inducing set $Z_1$ can be a subset of $X_1$ or it can contain pseudo inputs (Snelson & Ghahramani, 2006), i.e. points lying between the training inputs. For simplicity next we consider $Z_1$ as pseudo points, although in practice for continual learning it can be more suitable to select them from the training inputs (see Section 2.4). By evaluating the function output at each $z_{1,j}$ we obtain a vector of auxiliary function values $\mathbf{u}_1 = \{u_{1,j}\}_{j=1}^{M_1}$, where each $u_{1,j} = f(z_{1,j})$. Hence, we obtain the joint distribution

$$p(\mathbf{y}_1, \mathbf{f}_1, \mathbf{u}_1) = p(\mathbf{y}_1 | \mathbf{f}_1) p_\theta(\mathbf{f}_1 | \mathbf{u}_1) p_\theta(\mathbf{u}_1). \tag{2}$$

The exact posterior distribution $p_\theta(\mathbf{f}_1|\mathbf{u}_1, \mathbf{y}_1)p_\theta(\mathbf{u}_1|\mathbf{y}_1)$ is approximated by a distribution of the form, $q(\mathbf{f}_1, \mathbf{u}_1) = p_\theta(\mathbf{f}_1|\mathbf{u}_1)q(\mathbf{u}_1)$, where $q(\mathbf{u}_i)$ is a variational distribution and $p_\theta(\mathbf{f}_1|\mathbf{u}_1)$ is the GP prior conditional, $p_\theta(\mathbf{f}_1|\mathbf{u}_1) = \mathcal{N}(\mathbf{f}_1|K_{X_1 Z_1}K_{Z_1}^{-1}\mathbf{u}_1, K_{X_1} - K_{X_1 Z_1}K_{Z_1}^{-1}K_{Z_1 X_1})$. Here, $K_{X_1 Z_1}$ is the cross kernel matrix between the sets $X_1$ and $Z_1$, $K_{Z_1 X_1} = K_{X_1 Z_1}^\top$ and $K_{Z_1}$ is the kernel matrix on $Z_1$. The method learns $(q(\mathbf{u}_1), Z_1)$ by minimising the KL divergence $\text{KL}(p_\theta(\mathbf{f}_1|\mathbf{u}_1)q(\mathbf{u}_1)||p_\theta(\mathbf{f}_1|\mathbf{u}_1, \mathbf{y}_1)p_\theta(\mathbf{u}_1|\mathbf{y}_1))$. The ELBO is also maximised over the neural network feature vector parameters $\theta$ that determine the kernel matrices. This ELBO is generally written in the form (Hensman et al., 2013; Lloyd et al., 2015; Dezfouli & Bonilla, 2015; Hensman et al., 2015; Sheth et al., 2015),

$$\mathcal{F}(\theta, q(\mathbf{u}_1)) = \sum_{j=1}^{N_1} \mathbb{E}_{q(f_{1,j})}[\log p(y_{1,j}|f_{1,j})] - \text{KL}(q(\mathbf{u}_1)||p_\theta(\mathbf{u}_1)), \tag{3}$$

where $q(f_{1,j}) = \int p(f_{1,j}|\mathbf{u}_1)q(\mathbf{u}_1)d\mathbf{u}_1$ is an univariate Gaussian distribution with analytic mean and variance that depend on $(\theta, Z_1, q(\mathbf{u}_1), x_{1,j})$. Each expectation $\mathbb{E}_{q(f_{1,j})}[\log p(y_{1,j}|f_{1,j})]$ is a one-dimensional integral and can be estimated by Gaussian quadrature. The variational distribution $q(\mathbf{u}_1)$ is chosen to be a Gaussian, parameterised as $q(\mathbf{u}_1) = \mathcal{N}(\mathbf{u}_1|\mu_{u_1}, L_{u_1}L_{u_1}^\top)$, where $L_{u_1}$ is a square root matrix such as a lower triangular Cholesky factor. Then, based on the above we can jointly apply stochastic variational inference (Hensman et al., 2013) to maximise the ELBO over $(\theta, \mu_{u_1}, L_{u_1})$ and optionally over the inducing inputs $Z_1$.

## 2.2 LEARNING THE SECOND AND SUBSEQUENT TASKS

The functional regularisation framework for continual learning arises from the variational sparse GP inference method as we encounter the second and subsequent tasks.

Once we have learned the first task we throw away the dataset $(X_1, \mathbf{y}_1)$ and we keep in memory only a task summary consisting of the inducing inputs $Z_1$ and the variational Gaussian distribution $q(\mathbf{u}_1)$ (i.e. its parameters $\mu_{u_1}$ and $L_{u_1}$). Note also that $\theta$ (that determines the neural network feature vector $\phi(x; \theta)$) has a current value obtained by learning the first task. When the dataset $(X_2, \mathbf{y}_2)$ for the second task arrives, a suitable ELBO to continue learning $\theta$ and also estimate the second task summary $(Z_2, q(\mathbf{u}_2))$ is

$$\sum_{j=1}^{N_1} \mathbb{E}_{q(f_{1,j})}[\log p(y_{1j}|f_{1,j})] + \sum_{j=1}^{N_2} \mathbb{E}_{q(f_{2,j})}[\log p(y_{2,j}|f_{2,j})] - \sum_{i=1,2} \text{KL}(q(\mathbf{u}_i)||p_\theta(\mathbf{u}_i)),$$

which is just the sum of the corresponding ELBOs for the two tasks. We need to approximate this ideal objective by making use of the fixed summary $(Z_1, q(\mathbf{u}_1))$ that we have kept in memory for the first task. By considering $Z_1 \subset X_1$ as our replay buffer with outputs $\widetilde{\mathbf{y}}_1 \subset \mathbf{y}$ and $\widetilde{\mathbf{u}}_1 \subset \mathbf{f}_1$ the above can be approximated by

$$\frac{N_1}{M_1}\sum_{j=1}^{M_1} \mathbb{E}_{q(u_{1,j})}[\log p(\widetilde{y}_{1,j}|u_{1,j})] + \sum_{j=1}^{N_2} \mathbb{E}_{q(f_{2,j})}[\log p(y_{2,j}|f_{2,j}) - \sum_{i=1,2} \text{KL}(q(\mathbf{u}_i)|p_\theta(\mathbf{u}_i)),$$

where each $q(u_{1,j})$ is a univariate marginal of $q(\mathbf{u}_1)$. However, since $q(\mathbf{u}_1)$ is kept fixed the whole expected log-likelihood term $\frac{N_1}{M_1}\sum_{j=1}^{M_1} E_{q(u_{1,j})}[\log p(\widetilde{y}_{1,j}|u_{1,j})]$ is just a constant that does not depend on the parameters $\theta$ any more. Thus, the objective function when learning the second task reduces to maximising,

$$\mathcal{F}(\theta, q(\mathbf{u}_2)) = \sum_{j=1}^{N_2} \mathbb{E}_{q(f_{2,j})}[\log p(y_{2,j}|f_{2,j})] - \sum_{i=1,2} \text{KL}(q(\mathbf{u}_i)||p_\theta(\mathbf{u}_i)).$$

The only term associated with the first task is $\text{KL}(q(\mathbf{u}_1)||p_\theta(\mathbf{u}_1))$. While $q(\mathbf{u}_1)$ is fixed (i.e. its parameters are constant), the GP prior $p_\theta(\mathbf{u}_1) = \mathcal{N}(\mathbf{u}_1|\mathbf{0}, K_{Z_1})$ is still a function of the feature vector parameters $\theta$, since $K_{Z_1}$ depends on $\theta$. Thus, this KL term regularises the parameters $\theta$ so that, while learning the second task, the feature vector still needs to explain the posterior distribution over the function values $\mathbf{u}_1$ at input locations $Z_1$. Notice that $-\text{KL}(q(\mathbf{u}_i)||p_\theta(\mathbf{u}_i)$ is further simplified as $\int q(\mathbf{u}_1)\log p_\theta(\mathbf{u}_1)d\mathbf{u}_1 + const$, which shows that the regularisation is such that $p_\theta(\mathbf{u}_1)$ needs to be consistent with all infinite draws from $q(\mathbf{u}_1)$ in a moment-matching or maximum likelihood sense.

Similarly for the subsequent tasks we can conclude that for any new task $k$ the objective will be

$$\mathcal{F}(\theta, q(\mathbf{u}_k)) = \underbrace{\sum_{j=1}^{N_k} \mathbb{E}_{q(f_{k,j})} \log p(y_{k,j}|f_{k,j}) - \text{KL}(q(\mathbf{u}_k)||p_\theta(\mathbf{u}_k))}_{\text{objective for the current task}} - \underbrace{\sum_{i=1}^{k-1} \text{KL}(q(\mathbf{u}_i)||p_\theta(\mathbf{u}_i))}_{\text{regularisation from previous tasks}} . \quad (4)$$

Thus, functional regularisation when learning a new task is achieved through the sum of the KL divergences $\sum_{i=1}^{k-1} \text{KL}(q(\mathbf{u}_i)||p_\theta(\mathbf{u}_i))$ of all previous tasks, where each $q(\mathbf{u}_i)$ is the fixed posterior distribution which encodes our previously obtained knowledge about task $i < k$. Furthermore, in order to keep the optimisation scalable over tasks, we can form unbiased approximations of this latter sum by sub-sampling the KL terms, i.e. by performing minibatch-based stochastic approximation over the regularisation terms associated with these previous tasks.

## 2.3 Accurate weight space inference for the current task

While the above framework arises by applying sparse GP inference, it can still be limited. When the budget of inducing variables is small, the sparse GP approximation may lead to inaccurate estimates of the posterior belief $q(\mathbf{u}_k)$, which will degrade the quality of regularisation when learning new tasks. This is worrisome as in continual learning it is desirable to keep the size of the inducing set as small as possible.

One way to deal with this issue is to use a much larger set of inducing points for the current task or even maximise the full GP ELBO $\sum_{j=1}^{N_k} \mathbb{E}_{q(f_{k,j})} \log p(y_{k,j}|f_{k,j}) - \text{KL}(q(\mathbf{f}_k)||p_\theta(\mathbf{f}_k))$ (i.e. by using as many inducing points as training examples), and once training is completed to distill the small subset $Z_k, \mathbf{u}_k \subset X_k, \mathbf{f}_k$, and the corresponding marginal distribution $q(\mathbf{u}_k)$ from $q(\mathbf{f}_k)$, for subsequently regularising continual learning. However, carrying out this maximisation in the function space can be extremely slow since it scales as $O(N_k^3)$ per optimisation step. To our rescue, there is an alternative computationally efficient way to achieve this, by relying on the linear form of the kernel, that performs inference over the current task in the weight space. While this inference does not immediately provides us with the summary (induced points) for building the functional regularisation term, we can distill this term afterwards as discussed next. This allows us to address the continual learning aspect of the problem. Given that the current $k$-th task is represented in the weight space as $f_k(x; w_k) = w_k^\top \phi(x; \theta), w_k \sim \mathcal{N}(0, \sigma_w^2 I)$, we introduce a full Gaussian variational approximation $q(w_k) = \mathcal{N}(w_k|\mu_{w_k}, \Sigma_{w_k})$, where $\mu_k$ is a $K$ dimensional mean vector and $\Sigma_{w_k}$ is the corresponding $K \times K$ full covariance matrix parameterised as $\Sigma_{w_k} = L_{w_k} L_{w_k}^\top$. Learning the $k$-th task is carried out by maximising the objective in equation 4, with the only difference that the ELBO for the current task is now in the weight space. The objective becomes

$$\mathcal{F}(\theta, q(w_k)) = \sum_{j=1}^{N_k} \mathbb{E}_{q(w_k)}[\log p(y_{k,j}|w_k^\top \phi(x_{k,j}; \theta))] - \text{KL}(q(w_k)||p(w_k)) - \sum_{i=1}^{k-1} \text{KL}(q(\mathbf{u}_i)||p_\theta(\mathbf{u}_i)),$$

where $\mathbb{E}_{q(w_k)}[\log p(y_{k,j}|w_k^\top \phi(x_{k,j}, \theta))]$ can be re-written as one-dimensional integral and estimated using Gaussian quadrature. Once the variational distribution $q(w_k)$ has been optimised, together with the constantly updated feature parameters $\theta$, we can rely on this solution *to select inducing points $Z_k$*. See Section 2.4 for more detail. We also compute the posterior distribution over their function values $\mathbf{u}_k$ according to $q(\mathbf{u}_k) = \mathcal{N}(\mathbf{u}_k|\mu_{u_k}, L_{u_k} L_{u_k}^\top)$, where

$$\mu_{u_k} = \Phi_{Z_k} \mu_{w_k}, \quad L_{u_k} = \Phi_{Z_k} L_{w_k} \quad (5)$$

and the matrix $\Phi_{Z_k}$ stores as rows the feature vectors evaluated at $Z_k$. Subsequently, we store the $k$-th task summary $(Z_k, \mu_{u_k}, L_{u_k})$ and use it for regularising continual learning of subsequent tasks, by always maximising the objective $\mathcal{F}(\theta, q(w_k))$. Pseudo-code of the procedure is given in Algorithm 1.

## 2.4 Selection of the inducing points

After having seen the $k$-th task, and given that it is straightforward to compute the posterior distribution $q(\mathbf{u}_k)$ for any set of function values, the only issue remaining is to select the inducing inputs $Z_k$. A simple choice that works well in practice is to select $Z_k$ as a random subset of the training inputs $X_k$. The question is whether we can do better with some more structured criterion.

---

**Algorithm 1** Functional Regularised Continual Learning (FRCL) with task boundary detection

---

**Input:** Feature vector $\phi(x; \theta)$ with initial value of $\theta$, task $k = 0$, starting_time(k) = 10. Construct output weights $w_0$ and initialise variational parameters $\mu_{w_0}$ (e.g. around zero) and $L_{w_k} = I$.
  **for** $t = 1, 2, \ldots,$ **do**
    Receive next data minibatch $(X_t, \mathbf{y}_t)$.
    Compute KL values $\ell_t = \{\ell_{t,i}\}$, for any $x_{t,i} \in X_t$.
    **if** t - starting_time(k) > min_time_in(k) and StatisticalTest($\ell_t, \ell_{old}$) is significant **then**
      # A new task has been detected.
      Select inducing inputs $Z_k$ for current task.
      Compute the parameters of $q(\mathbf{u}_k)$ from equation 5 and store them.
      Construct new output weights $w_{k+1}$ and variational parameters $(\mu_{w_{k+1}}, L_{w_{k+1}})$.
      k = k + 1; starting_time(k) = t.
    **else**
      $\ell_{old} = \ell_t$.
    **end if**
    Gradient step to update $(\theta, \mu_{w_k}, L_{w_k})$ by maximising $\mathcal{F}(\theta, q(w_k))$.
  **end for**

---

In our experiments we will investigate several criteria where the most effective one will be an unsupervised criterion that only depends on the training inputs, while the other supervised criteria are described in the Appendix. This unsupervised criterion quantifies how well we reconstruct the full kernel matrix $K_{X_k}$ from the inducing set $Z_k$ and it can be expressed as the trace of the covariance matrix of the prior GP conditional $p(\mathbf{f}_k | \mathbf{u}_k)$, i.e.

$$\mathcal{T}(Z_k) = \text{tr}\left(K_{X_k} - K_{X_k Z_K} K_{Z_k}^{-1} K_{Z_k X_k}\right) = \sum_{j=1}^{N_k} \left[k\left(x_{k,j}, x_{k,j}\right) - k_{Z_K, x_{k,j}}^{\top} K_{Z_k}^{-1} k_{Z_k, x_{k,j}}\right], \quad (6)$$

where each $k(x_{k,j}, x_{k,j}) - k_{Z_K, x_{k,j}}^{\top} K_{Z_k}^{-1} k_{Z_k, x_{k,j}} \geq 0$ is a reconstruction error for an individual training point. The above quantity appears in the ELBO in (Titsias, 2009), is also used in (Csato & Opper, 2002) and it has deep connections with the Nyström approximation (Williams & Seeger, 2001) and principal component analysis. The criterion in equation 6 promotes finding inducing points $Z_k$ that are repulsive with one another and are spread evenly in the input space under a similarity/distance implied by the dot product of the feature vector $\phi(x; \theta_k)$ (with $\theta_k$ being the current parameter values after having trained with the $k$-th task). An illustration of this repulsive property is given in Section 4.

To select $Z_k$, we minimise $\mathcal{T}(Z_k)$ by applying discrete optimisation where we select points from the training inputs $X_k$. The specific optimisation strategy we use in the experiments is to start with an initial random set $Z_k \subset X_k$ and then further refine it by doing local moves where random points in $Z_k$ are proposed to be changed with random points of $X_k$.

## 2.5 PREDICTION AND DIFFERENCES WITH WEIGHT SPACE METHODS

Prediction at any $i$-th task that has been encountered in the past follows the standard sparse GP predictive equations. Given a test data point $x_{i,*}$ the predictive density of its output value $y_{i,*}$ takes the form $p(y_{i,*}) = \int p(y_{i,*}|f_{i,*})p_\theta(f_{i,*}|\mathbf{u}_i)q(\mathbf{u}_i)d\mathbf{u}_i df_{i,*} = \int p(y_{i,*}|f_{i,*})q_\theta(f_{i,*})df_{i,*}$ where $q_\theta(f_{i,*}) = \mathcal{N}(f_{i,*}|\mu_{i,*}, \sigma_{i,*}^2)$ is an univariate posterior Gaussian with mean and variance,

$$\mu_{i,*} = \mu_{u_i}^{\top} K_{Z_i}^{-1} \Phi_{Z_1} \phi(x_{i,*}; \theta), \quad \sigma_{i,*}^2 = k(x_{i,*}, x_{i,*}) + k_{Z_i x_{i,*}}^{\top} K_{Z_i}^{-1} \left[L_{u_i} L_{u_i}^{\top} - K_{Z_i}\right] K_{Z_i}^{-1} k_{Z_i x_{i,*}},$$

where in $\mu_{i,*}$ we have explicitly written the cross kernel vector $k_{Z_k x_{i,*}} = \Phi_{Z_i} \phi(x_{i,*}; \theta)$ (assuming $\sigma_w^2 = 1$ for simplicity) to reveal a crucial property of this prediction. Specifically, given that $f_{i,*} = w_i^{\top} \phi(x_{i,*}; \theta)$ the vector $\mu_{u_i}^{\top} K_{Z_i}^{-1} \Phi_{Z_i}$ acts as a mean prediction for the task-specific parameter row vector $w_i^{\top}$. As we learn future tasks and the parameter $\theta$ changes, this mean parameter prediction automatically adapts (recall that $K_{Z_i} = \Phi_{Z_i} \Phi_{Z_i}^{\top}$ and $\Phi_{Z_i}$ vary with $\theta$ and only $\mu_{u_i}$ is constant) in order to counteract changes in the feature vector $\phi(x_{i,*}; \theta)$, so that the overall prediction for the function value, i.e. $\mu_{i,*} = \mathbb{E}[f_{i,*}]$, does not become obsolete. For instance, the prediction of the function values at the inducing inputs $Z_i$ always remains constant to our fixed/stored mean belief $\mu_{u_i}$

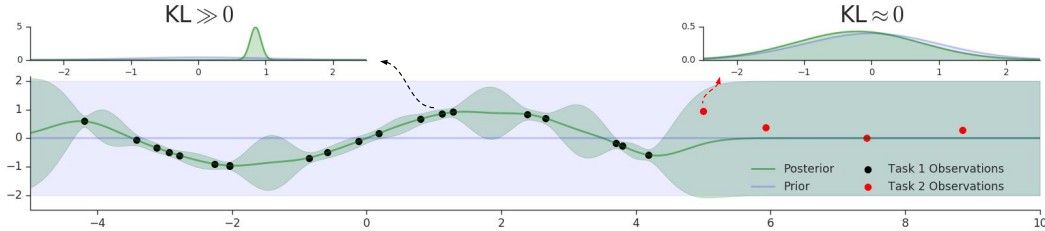

**Figure 2:** Detecting task boundaries using the predictive uncertainty of a Gaussian Process. As GP predictions revert to the prior (shaded blue) when queried far from observed data (shown as black dots), we can test for a distribution shift by comparing the GP posterior over functions (in green) to the prior. Small distance between predictive distributions at test points (red dots) suggest a task switch.

since by setting $x_{i,*} = Z_i$ the formula gives $\mu_{u_i}^\top K_{Z_i}^{-1} \Phi_{Z_i} \Phi_{Z_i}^\top = \mu_{u_i}^\top$. Similar observations can be made for the predictive variances.

The above analysis reveals an important difference between continual learning in function space and in weight space, where in the latter framework task-specific parameters such as $w_i$ might not automatically adapt to counteract changes in the feature vector $\phi(x; \theta)$, as we learn new tasks and $\theta$ changes. For instance, if as a summary of the task, instead of the function space posterior distribution $q(\mathbf{u}_i)$, we had kept in memory the weight space posterior $q(w_i)$ (see Section 2.3), then the corresponding mean prediction on the function value, $\mathbb{E}[f_{i,*}] = \mu_{w_i}^\top \phi(x_{i,*}; \theta)$, can become obsolete as $\phi(x_{i,*}; \theta)$ changes and $\mu_{w_i}$ remains constant.

## 3 DETECTING TASK BOUNDARIES USING BAYESIAN UNCERTAINTIES

So far we have made the assumption that task switches are known, which may not always be a realistic setting. Instead, we now introduce a novel approach for detecting task boundaries in continual learning arising naturally from our method by a simple observation: The GP predictive uncertainty grows as the model is queried far away from observed data, eventually falling back to the prior. When a minibatch of data $\{x_i, y_i\}_{i=1}^b$ from a new task arrives, we thus expect the distance between prior and posterior to be small (see Figure 2). Thus, a simple way to detect a change in the input distribution is to compare the GP univariate posterior density

$$q(f(x_i)) = \mathcal{N}(f(x_i)|\mu_i, \sigma_i^2) \approx \int p(f(x_i)|\mathbf{f})p(\mathbf{f}|\mathbf{y}, X)df(x_i),$$

where $(\mu_i, \sigma_i^2)$ are predictive mean and variance, with the prior GP density $p(f(x_i)) = \mathcal{N}(f(x_i)|0, k(x_i, x_i))$. This can be achieved by using a divergence measure between distributions such as the symmetrised KL divergence,

$$\ell_i = 0.5\Big( \int q(f(x_i)) \log \frac{q(f(x_i))}{p(f(x_i))} df(x_i) + \int p(f(x_i)) \log \frac{p(f(x_i))}{q(f(x_i))} df(x_i)\Big), \; i = 1, \ldots, b,$$

computed separately for any $x_i$ in the minibatch. Given that all distributions are univariate Gaussians the above can be obtained analytically. When each score $\ell_i$ is close to zero this indicate that the input distribution has changed so that a task switch can be detected. Each $\ell_i \geq 0$ can be thought of as expressing a degree of surprise about $x_i$, i.e. the smaller is $\ell_i$ the more surprising is $x_i$. Thus our idea has close links to Bayesian surprise (Itti & Baldi, 2006).

In order to use this intuition to detect task switches, we can perform a statistical test between the values $\{\ell_i\}_{i=1}^b$ for the current batch and those from the previous batch $\{\ell_i^{old}\}_{i=1}^b$ before making any updates to the parameters using the current batch. A suitable choice is Welch's t-test (due to unequal variances), demanding that with high statistical significance the mean of $\{\ell_i\}_{i=1}^b$ is smaller than the mean of $\{\ell_i^{old}\}_{i=1}^b$.

The ability to detect changes based on the above procedure arises from our framework as we construct posterior distributions over function values $f(x_i)$ that depend on inputs $x_i$ (while in contrast a posterior over weights alone does not depend on any input). Subsequently, these predictive densities contain information about the distribution of these inputs in the sense that when an $x_i$ is close to the

training inputs from the same task we expect reduced uncertainty, while for inputs of a different task we expect high uncertainty that falls back to the prior uncertainty.

# 4 EXPERIMENTS

We now test the scalability and competitiveness of our method on various continual learning problems, referring to the proposed approach as Functional Regularised Continual Learning (FRCL). Throughout this section, we will aim to answer three key questions:

  **(i)** How does FRCL compare to state-of-the-art algorithms for Continual Learning?

  **(ii)** How important is a principled criterion for inducing point selection? How do varying numbers of inducing points/task affect overall performance?

  **(iii)** If ground truth task boundaries are not given, does the detection method outlined in Section 3 succeed in detecting task changes?

In order to answer these questions, we consider experiments on three established Continual Learning classification problems: Split-MNIST, Permuted-MNIST and sequential Omniglot (Goodfellow et al., 2013; Zenke et al., 2017; Schwarz et al., 2018), described in the Appendix. FRCL methods have been implemented using GPflow (Matthews et al., 2017).

In addition to comparing our method with other approaches in the literature by quoting published results, we also show results for an additional baseline (BASELINE) corresponding to a simple replay-buffer method for Continual Learning (explained in the Appendix).

## 4.1 IS FRCL A COMPETITIVE MODEL FOR CONTINUAL LEARNING?

Addressing first question (i), we show results on the MNIST-variations in Table 1 and on the more challenging Omniglot benchmark in Table 2. Note that we also specify the inducing point optimisation criterion in brackets, i.e. FRCL (TRACE-TERM) corresponds to the loss in equation 6. We observe strong results on all benchmarks, setting a new state-of-the-art results on Permuted-MNIST & Omniglot while coming close to existing results on Split-MNIST. The improvement on the BASELINE shows that approximate posterior distributions over functions values can lead to more effective regularisation for CL compared to just having a replay buffer of input-output pairs. Furthermore, despite its simplicity, the simple BASELINE strategy performs competitively. In line with other results, we conclude that rehearsal-based ideas continue to provide a strong baseline. This also gives justification to a main motivation of this work:

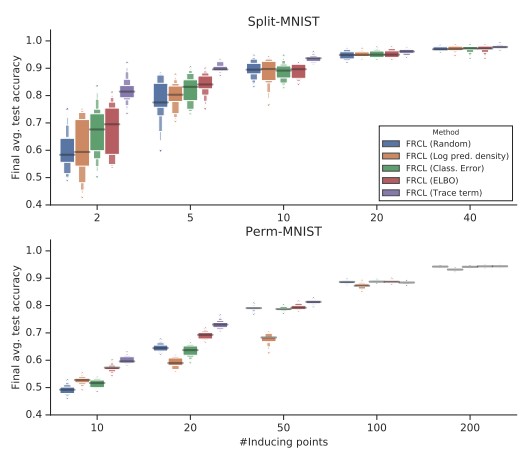

**Figure 3:** Comparing optimisation criteria for varying number of inducing points.

To unite the two previously separate lines of CL work on rehearsal-based and Bayesian methods. Nevertheless, other methods may be more suitable when storing data is not feasible.

## 4.2 INDUCING POINT OPTIMISATION AND TASK SWITCH DETECTION

An appealing theoretical property of our method is the principled selection of inducing points through optimisation. Answering question (ii), we now proceed to investigate the importance of the criterion used as well as the dependence on the number of inducing points. These results are shown in Figure 3. Note that the definition of objectives *Class. Error, ELBO & Log pred. density* are given in the Appendix. In accordance with our intuition, we observe that optimisation becomes increasingly important as the number of inducing points is reduced. The results also give strong statistical motivation to use the trace-term motivated before. Further, as can be seen looking at the results for *Log pred. density*, a poorly chosen criterion may behave worse than random.

**Table 1:** Results on Permuted- and Split-MNIST. Baseline results are taken from Nguyen et al. (2017). For the experiments conducted in this work we show the mean and standard deviation over 10 random repetitions. Where applicable, we also report the number of inducing points/replay buffer size per task in parentheses.

| Algorithm | Permuted-MNIST | Split-MNIST |
|---|---|---|
| DLP (Eskin et al., 2004) | 82% | 61.2% |
| EWC (Kirkpatrick et al., 2017) | 84% | 63.1% |
| SI (Zenke et al., 2017) | 86% | **98.9%** |
| VCL (Nguyen et al., 2017) | 90% | 97.0% |
| + random Coreset | 93% (200 points/task) | |
| + k-center Coreset | 93% (200 points/task) | |
| + unspecified Coreset selection | | 98.4% (40 points/task) |
| BASELINE | 48.6% $\pm$ 1.7 (10 points/task) | |
| FRCL (RANDOM) | 48.2% $\pm$ 4.0 (10 points/task) | 59.8% $\pm$ 8.0 (2 points/task) |
| FRCL (TRACE) | 61.7% $\pm$ 1.8 (10 points/task) | 82.0% $\pm$ 5.0 (2 points/task) |
| BASELINE | 82.3% $\pm$ 0.3 (200 points/task) | 95.8% $\pm$ 1.1 (40 points/task) |
| FRCL (RANDOM) | 94.2% $\pm$ 0.1 (200 points/task) | 97.1% $\pm$ 0.7 (40 points/task) |
| FRCL (TRACE) | **94.3**% $\pm$ 0.2 (200 points/task) | **97.8**% $\pm$ 0.7 (40 points/task) |

**Table 2:** Results on sequential Omniglot. Baseline results are taken from Schwarz et al. (2018). Shown are mean and standard deviation over 5 random task permutations. Note that methods *'Single model per Task'* and *'Progressive Nets'* are not directly comparable due to unrealistic assumptions, but serve as an upper bound on the performance for the remaining continual learning methods.

| Algorithm | Test Accuracy | | |
|---|---|---|---|
| Single model per Task | 88.34 | | |
| Progressive Nets | 86.50 $\pm$ 0.9 | | |
| Finetuning | 26.20 $\pm$ 4.6 | | |
| Learning Without Forgetting | 62.06 $\pm$ 2.0 | | |
| Elastic Weight Consolidation (EWC) | 67.32 $\pm$ 4.7 | | |
| Online EWC | 69.99 $\pm$ 3.2 | | |
| Progress & Compress | 70.32 $\pm$ 3.3 | | |
| **Methods evaluated in this paper** | **1 point/char** | **2 points/char** | **3 points/char** |
| BASELINE | 42.73 $\pm$ 1.2 | 57.17 $\pm$ 1.2 | 65.32 $\pm$ 1.1 |
| FRCL (RANDOM) | 69.74 $\pm$ 1.1 | 80.32 $\pm$ 2.5 | **81.42** $\pm$ 1.2 |
| FRCL (TRACE) | **72.02** $\pm$ 1.3 | **81.47** $\pm$ 1.6 | 81.01 $\pm$ 1.1 |

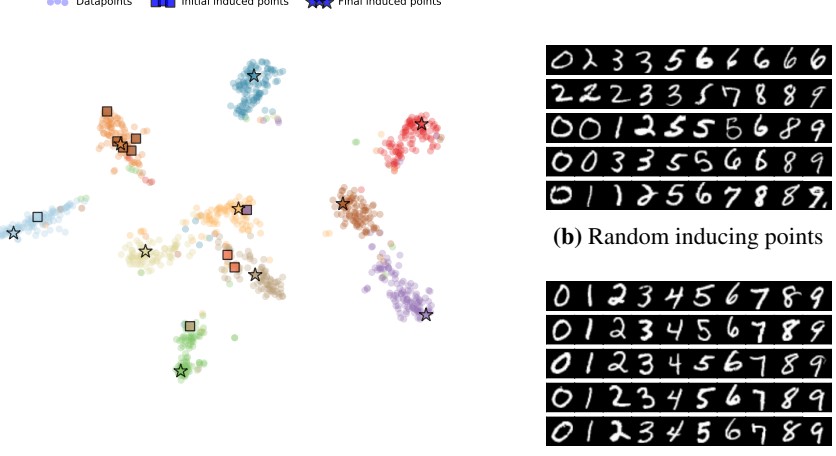

(a) A TSNE projection of inducing points in feature space.  (b) Random inducing points  (c) Optimised inducing points

**Figure 4:** Inducing point optimisation for the first task on the Permuted-MNIST benchmark. The number of inducing points was limited to 10. **Left:** A example optimisation shown in the feature space of a trained network. Points are coloured by class label. Data shown corresponds to the first row in the images on the right. **Right:** Optimised inducing points consistently cover examples of all classes. Each row corresponds to a different run with random initialisation. Best viewed in colour.

To provide an insight into the solutions obtained by the trace-term criterion, we provide a visualisation of inducing points in Figure 4. Remarkably, even though the objective is unsupervised, it results in a consistent allocation of one example per class. Furthermore, the optimised inducing points are spread across the input space as shown by the TNSE (Maaten & Hinton, 2008) visualisation, which is in line with the intuition that the objective encourages repulsive inducing points.

Finally, we answer question (iii) by first showing both the mean of the terms $\{\ell_i\}_{i=1}^b$ (top) as well as the result of Welch's t-test (bottom) between terms $\{\ell_i\}_{i=1}^b$, $\{\ell_i^{old}\}_{i=1}^b$ in Figure 5, using only a small number Omniglot alphabets and 1000 training iterations per task for illustrative purposes. We note that the intuition built up in Section 3 holds, with clear spikes being shown whenever the t-test returns a positive result.

Furthermore, we provide a quantitative comparison in Table 3. On the positive side, we note very strong results for Split- & Perm-MNIST and further observe that we find a similar t-test threshold value to apply to all dataset, making this an easy hyper-parameter to set. While the task boundary detection results for Omniglot are less strong, which may due to the smaller batch size (32 for Omniglot, $\geq 100$ for the MNIST-versions), resulting in a noisier test result. Note that this could be easily mitigated by using a larger set $\{\ell_i^{old}\}$, e.g. the last 10 minibatches, which would make this test more robust.

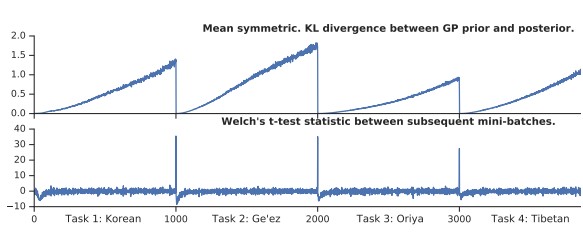

**Figure 5:** Visualising KL terms and test statistics on multiple Omniglot tasks.

**Table 3:** Task boundary detection evaluated as a binary classification task (Positive labels corresponds to task switches).

| | Precision/Recall/F1 @ Threshold | | |
|---|---|---|---|
| | Split-MNIST | Perm-MNIST | Omniglot |
| 4.0 | 0.94/1.0/0.97 | 0.43/0.98/0.60 | 0.13/0.90/0.23 |
| 5.0 | **1.0/1.0/1.0** | 0.95/0.94/0.94 | **0.71/0.75/0.73** |
| 6.0 | **1.0/1.0/1.0** | **1.0/0.91/0.95** | 0.89/0.59/0.71 |
| 7.0 | **1.0/1.0/1.0** | **1.0/0.91/0.95** | 0.89/0.46/0.60 |
| 8.0 | **1.0/1.0/1.0** | 1.0/0.86/0.92 | 0.88/0.32/0.46 |

## 5 DISCUSSION

We introduced a functional regularisation approach for supervised continual learning that combines inducing point GP inference with deep neural networks. Our method constructs task-specific posterior beliefs or summaries on inducing inputs. Subsequently, the task-specific summaries allow us to regularise continual learning and avoid catastrophic forgetting. Our approach unifies the two extant approaches to continual learning, of parameter regularisation and replay/rehersal. Viewed from the regularisation perspective, our approach regularises the functional outputs of the neural network, thus avoid the brittleness due to representation drift. Viewed from a rehearsal method perspective, we provide a principled way of compressing data from previous task, by means of optimizing the selection of inducing points. By investigating the behaviour of the posterior beliefs, we also proposed a method for detecting task boundaries. All these improvements lead to strong empirical gains compared to state-of-the-art continual learning methods.

Regarding related work on online learning using GPs, notice that previous algorithms (Bui et al., 2017a; Csato & Opper, 2002; Csato, 2002) learn in an online fashion a single task where data from this task arrive sequentially. In contrast in this paper we developed a continual learning method for dealing with a sequence of different tasks.

A direction for future research is to enforce a fixed memory buffer (or a buffer that grows sub-linearly w.r.t. the number of tasks), in which case one would need to compress the summaries of all previous seen tasks into a single summary. Finally, while in this paper we applied the method to supervised classification tasks, it will be interesting to consider also applications in other domains such as reinforcement learning.

### ACKNOWLEDGEMENTS

We would like to thank Hyunjik Kim and Raia Hadsell for useful discussions and feedback.

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

## A  TASK DESCRIPTIONS

**Split-MNIST and Permuted-MNIST.** Among a large number of diverse experiments in continual learning publications, two versions of the popular MNIST dataset have recently started to become increasingly popular benchmarks: Permuted- and Split-MNIST. In Permuted-MNIST (e.g. Goodfellow et al., 2013; Kirkpatrick et al., 2017; Zenke et al., 2017), each task is a variant of the initial 10-class MNIST classification task where all input pixels have undergone a fixed (random) permutation. The Split-MNIST experiment was introduced by Zenke et al. (2017): Five binary classification tasks are constructed from the classes in the following order: 0/1, 2/3, 4/5, 6/7, and 8/9.

**Omniglot.** To assess our method under more challenging conditions, we consider the sequential Omniglot task proposed for continual learning in Schwarz et al. (2018). Omniglot Lake et al. (2011) is a dataset of 50 alphabets, each with a varying number of classes/characters which we treat of as a sequence of distinct classification problems. As suggested in Schwarz et al. (2018), for Omniglot, we apply data-augmentation and use the same train/validation/test split. Following the same experimental setup proposed, we used an identical convolutional network to construct the feature vector $\phi(x; \theta)$. Results reported are obtained by training on the union of training and validation set after choosing any hyper-parameters based on the validation set. Note that all experiments were run with data processing and neural network construction code kindly provided by the authors in Schwarz et al. (2018), ensuring directly comparable results.

Given that Permuted-MNIST and Omniglot are multi-class classification problems, where each $k$-th task involves classification over $C_k$ classes, we need to generalise the model and the variational method to deal with multiple GP functions per task. This is outlined in the next section.

## B  EXTENSION TO MULTI-CLASS (OR MULTIPLE-OUTPUTS) TASKS

For multi-class classification problems, such as permuted MNIST and Omniglot considered in our experiments, where in general each $k$-th task involves classification over varying number of classes, we need to extend the method to deal with multiple functions per task. For instance, assume that the $k$-th task is a multi-class classification problem that inolves $C_k$ classes. To model this we need $C_k$ independent draws from the GP, such that $f_k^c(x) \sim \mathcal{GP}(0, k(x, x'))$ with $c = 1, \ldots, C_k$, which are combined based on a multi-class likelihood such as softmax. While in the main text we presented the method assuming a single GP function per task, the generalisation to multiple functions is straightforward by assuming that all variational approximations factorise across different functions/classes. For example, the variational distribution over all inducing variables $U_k = \{\mathbf{u}_k^c\}_{c=1}^{C_k}$ takes the form $q(U_k) = \prod_{c=1}^{C_k} q(\mathbf{u}_k^c)$ and similarly the variational approximation over the task weights $W_k = \{w_k^c\}$, needed in the ELBO in Section 3.2, also factorises across classes. Notice also that all inducing variables $U_k$ are evaluated on the same inputs $Z_k$. Furthermroe, the KL regularization term for each task takes the form of a sum over the different functions, i.e. $\mathrm{KL}(q(U_k)||p_\theta(U_k)) = \sum_{c=1}^{C_k} \mathrm{KL}(q(\mathbf{u}_k^c)||p_\theta(\mathbf{u}_k^c))$.

## C  BASELINE MODEL

The BASELINE model (see main text) is based on storing an explicit replay buffer $(\widetilde{\mathbf{y}}_i, \widetilde{X}_i)$, i.e. a subset of the training data where $\widetilde{\mathbf{y}}_i \subset \mathbf{y}_i$ and $\widetilde{X}_i \subset X_i$, for each past task. Then, at each step when we encounter the $k$-th task training is performed by optimising an unbiased estimate of the full loss

(i.e. if we had all $k$ tasks at once), given by

$$L(\theta, w_{1:k}) = \ell_k(\mathbf{y}_k, X_k; w_k, \theta) + \sum_{i=1}^{k-1} \frac{N_i}{M_i} \ell_i(\widetilde{\mathbf{y}}_i, \widetilde{X}_i; w_i, \theta),$$

where each $\ell_i(\cdot)$ is a task-specific loss function, such as cross entropy for classification, and each scalar $\frac{N_i}{M_i}$ corrects for the bias on the loss value caused by approximating the initial full loss by a random replay buffer of size $M_i$. All output weights $w_{i:k}$ of the current and old tasks, in the multi-head architecture, are constantly updated together with the feature parameter vector $\theta$. Also at each step a fresh set of output weights is constructed in order to deal with the current task.

## D    SELECTION OF THE INDUCING POINTS AND OPTIMISATION CRITERIA

After having seen the $k$-th task, and given that it is straightforward to compute the posterior distribution $q(\mathbf{u}_k)$ for any set of function values, the only issue remaining is to select the inducing inputs $Z_k$. A simple choice that works well in practice is to select $Z_k$ as a random subset of the training inputs $X_k$. The question is whether we can do better with some more structured criterion. In our experiments we will investigate three supervised criteria, that make use of class labels, and one unsupervised that only depends on inputs and the neural network feature vector. All criteria below optimise over $Z_k$ using discrete search.

The first supervised criterion is to minimize the negative average log predictive density,

$$\text{Log pred. density}(Z_k) = -\frac{1}{|X_k \setminus Z_k|} \sum_{x_{k,j} \in X_k \setminus Z_k} \log p(y_{k,j}|Z_k), \qquad (7)$$

computed at all remaining training inputs by excluding the selected inducing inputs $Z_k$. Each predictive density $p(y_{k,j}|Z_k)$ is obtained through the inducing inputs $Z_k$ and it takes the form

$$p(y_{k,j}|Z_k) = \int p(y_{k,j}|f_{k,j})q(\mathbf{u}_k|Z_k)d\mathbf{u}_k = \int p(y_{k,j}|f_{k,j})\mathcal{N}(\mathbf{u}_k|\mu_{u_k}, L_{u_k}L_{u_k}^\top)d\mathbf{u}_k, \quad (8)$$

which for the classification problems, where the likelihood $p(y_{k,j}|f_{k,j})$ is not Gaussian, is computed numerically by one-dimensional Gaussian quadrature. Notice that for the multi-class case the predictive density is obtained by integrating over $q(U_k) \equiv q(U_k|Z_k)$ (see previous Appendix B). Modern GP packages, such as GPflow, provide efficient implementation of the above predictive densities.

The second supervised criterion is similar, but $-\log p(y_{k,j}|Z_K)$ is replaced by classification error score $I(y_{k,j} \neq y_{k,j}^*)$,

$$\text{Class. Error}(Z_k) = \frac{1}{|X_k \setminus Z_k|} \sum_{x_{k,j} \in X_k \setminus Z_k} I(y_{k,j} \neq y_{k,j}^*), \qquad (9)$$

where $y_{k,j}^*$ is the predicted label, which is obtained by first computing the predictive density $p(y_{k,j}|Z_k)$ conditional on a given set of inducing inputs $Z_k$ and then choosing $y_{k,j}^*$ by taking argmax (i.e. selecting the class with the largest predictive probability).

Both criteria above essentially capture how well we predict the remaining points from the inducing points $Z_k$, thus good choices for $Z_k$ are presumably those that lead to good prediction for the remaining points.

The third supevised criterion is the standard sparse GP variational lower bound,

$$\text{ELBO}(Z_k) = \sum_{j=1}^{N_k} \mathbb{E}_{q(f_{k,j})}[\log p(y_{k,j}|f_{k,j})] - \text{KL}(q(\mathbf{u}_k|Z_k)||p_\theta(\mathbf{u}_k|Z_k)), \qquad (10)$$

which is viewed purely as a function of $Z_k$ while all remaining quantities are constant. This criterion tries to optimise over $Z_k$ in order to approximate as best as possible the marginal likelihood on the training data and it is the one used by all variational sparse GP training methods (although there the ELBO is maximized jointly over both $Z_k$ with the remaining parameters).

The last unsupervised criterion (also discussed in the main paper) expresses how well we reconstruct the full kernel matrix $K_{X_k}$ from the inducing set $Z_k$, which can be described by

$$\text{Trace term}(Z_k) = \left(K_{X_k} - K_{X_k Z_k} K_{Z_k}^{-1} K_{Z_k X_k}\right) = \sum_{j=1}^{N_k} \left[ k(x_{k,j}, x_{k,j}) - k_{Z_K, x_{j,k}}^{\top} K_{Z_k}^{-1} k_{Z_k, x_{i,k}} \right], \tag{11}$$

where each $k(x_{k,j}, x_{k,j}) - k_{Z_K, x_{j,k}}^{\top} K_{Z_k}^{-1} k_{Z_k, x_{i,k}} \geq 0$ is a reconstruction error for an individual training point. The above quantity appears in the ELBO in (Titsias, 2009), is also used in (Csato & Opper, 2002) and it has deep connections with the Nyström approximation (Williams & Seeger, 2001) and principal component analysis. The criterion in equation 6 promotes finding inducing points $Z_k$ that are repulsive with one another and are spread evenly in the input space under a similarity/distance implied by the dot product of the feature vector $\phi(x; \theta_k)$ (with $\theta_k$ being the current parameter values after having trained with the $k$-th task). An illustration of this is given in the Figure 4.

Figure 4 in the Experiments Section illustrates the optimisation of the inducing inputs in Permuted-MNIST. Also, Figure 6(a) shows randomly chosen (3 examples per class) for the Greek alphabet/task in the Omniglot experiments, while Figure 6(b) shows the corresponding points after have been optimised by using the trace criterion above.

Figure 3 shows the evolution of the performance for all selection criteria on individual tasks for Split-MNIST and Permuted-MNIST.

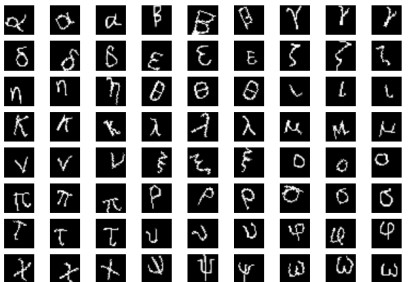
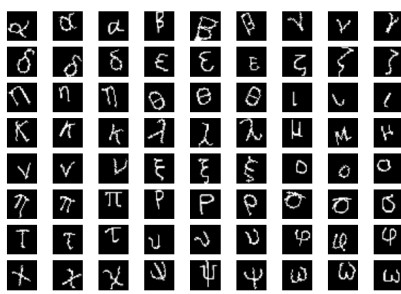

(a) Randomly selected inducing points          (b) Optimised inducing points

**Figure 6:** Inducing points for Greek alphabet of the Omniglot benchmark. The number of inducing points was limited to 3 per character.

# E  TASK BOUNDARY DETECTION

In order to apply the task-boundary detection for multiple classes, we perform Welch's t-test separately for each function. We consider mean, median and the maximum of those t-test results to make a decision. Furthermore, we also found that conducting the test in log-space significantly improved robustness of the test results and thus find that tests using the max over functions in log-space allow for higher thresholds and more robust results. In order to give justification to those conclusions we show our experiments on Permuted-MNIST in Figure 7, once again using 10 random task permutations.

Note also that after a task boundary was detected, we do not consider a new test for the next 10 iterations.

At this point it should be said that while the boundary detection method is simple, cheap to compute, unsupervised and general, it is not applicable when the Continual Learning faces a continuum of tasks without clear switches or when the input distribution is constant and different tasks merely correspond to varying labels.

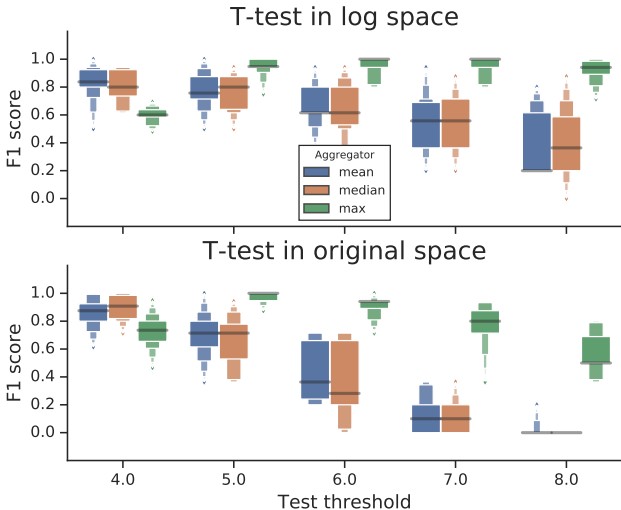

**Figure 7:** Ablation study comparing aggregation methods and whether tests should be conducted in log-space. Results are shown on Permuted MNIST, 10 random task permutations.

**Table 4:** Hyperparameters for the experiments on Split MNIST. Optimal values (in bold) were chosen on the validation set. Test set results were obtained by training on the union of training&validation set using those values.

| Parameter | Considered range | Comment |
|---|---|---|
| Network size (#Layers × Units) | $\{\mathbf{2}\} \times \{\mathbf{256}\}$ | Based on Zenke et al. (2017). |
| Activation function | $\{f(x) : \max(0, x) \text{ (ReLu)}\}$ | " |
| Learning rate | $\{\mathbf{5 \cdot 10^{-4}}, 10^{-4}, 5 \cdot 10^{-5}\}$ | |
| Batch size | $\{32, 64, \mathbf{100}, 128\}$ | |
| #Training steps | $\{2500, \mathbf{3000}, 3500\}$ | |
| #Inducing points optimisation steps | $\{\mathbf{1000}, 2000\}$ | No significant difference. |

## F  EXPERIMENTAL DETAILS

Experimental details for all experiments are shown in Tables 4, 5 and 6. Note that for the MNIST results, we obtain final results after optimising hyperparameters on the validation set and using those values to train on the union of training & validation set.

For Omniglot on the other hand, we report final test-set results only after training on the training set in order to remain consistent with the results in Schwarz et al. (2018)

**Table 5:** Hyperparameters for the experiments on Permuted MNIST. Optimal values (in bold) were chosen on the validation set. Test set results were obtained by training on the union of training&validation set using those values.

| Parameter | Considered range | Comment |
|---|---|---|
| Network size (#Layers × Units) | $\{\mathbf{2}\} \times \{\mathbf{100}\}$ | Based on Zenke et al. (2017). |
| Activation function | $\{f(x) : \max(0, x) \text{ (ReLu)}\}$ | " |
| Learning rate | $\{\mathbf{10^{-3}}, 5 \cdot 10^{-4}, 10^{-4}, 5 \cdot 10^{-5}\}$ | |
| Batch size | $\{32, 64, \mathbf{128}\}$ | |
| #Training steps | $\{\mathbf{2000}, 2500\}$ | No significant difference. |
| #Inducing points optimisation steps | $\{\mathbf{1000}, 2000\}$ | No significant difference. |

**Table 6:** Hyperparameters for the experiments on Omniglot. Optimal values (in bold) were chosen on the validation set. Test set results were obtained by training on the union of training&validation set using those values.

| Parameter | Considered range | Comment |
|---|---|---|
| Conv. filters | $\{[64, 64, 64, 64]\}$ | Based on Vinyals et al. (2016). |
| Conv. Kernel size | $\{3 \times 3\}$ | " |
| Conv. Padding | $\{SAME\}$ | " |
| Max Pool. Kernel size | $\{3 \times 3\}$ | " |
| Max Pool. stride | $\{2 \times 2\}$ | " |
| Max Pool. Padding | $\{VALID\}$ | " |
| Activation function | $\{f(x) : \max(0, x) \text{ (ReLu)}\}$ | " |
| Learning rate | $\{\mathbf{10^{-3}}, 5 \cdot 10^{-4}, 10^{-4}\}$ | |
| Batch size | $\{\mathbf{32}\}$ | |
| #Training steps | $\{\mathbf{2500}\}$ | |
| #Inducing points optimisation steps | $\{\mathbf{1000}, 2000\}$ | No significant difference. |

## G    COMPARISON TO VCL ON OMNIGLOT

To provide a further comparison to VCL (Nguyen et al., 2017), we show results on Omniglot using Multi-Layer Perceptrons (MLP). A comparison with MLPs is due to the fact that reliable variational inference methods for CNNs (which are usually used for Omniglot) are yet to be developed. All our results for VCL are obtained using code provided by the authors.

For all experiments, we used an MLP with 4 hidden layers of 256 units each and ReLU activations, a batch size of 100 and the Adam Optimiser (Step size of 0.001 for VCL and 0.0001 for FRCL). We optimised both types of algorithms independently and found the following parameters:

1. VCL: 100 training epochs per task, 50 adaptation epochs to coreset, Multi-Head
2. FRCL (TRACE): 1500 training steps per task, 2000 discrete optimization steps, inducing points initialised as a uniform distribution over classes.

**Table 7:** Results on sequential Omniglot using a MLP.

| Algorithm | Test Accuracy | | |
|---|---|---|---|
| VCL (NO CORESET) | $48.4 \pm 0.7$ | | |
| | **1 point/char** | **2 points/char** | **3 points/char** |
| VCL (RANDOM CORESET) | $49.18 \pm 2.1$ | $50.5 \pm 1.2$ | $51.64 \pm 1.0$ |
| VCL (K-CENTER CORESET) | $48.89 \pm 1.1$ | $49.58 \pm 1.4$ | $49.61 \pm 1.0$ |
| FRCL (TRACE) | $48.84 \pm 1.1$ | $52.10 \pm 1.2$ | $53.86 \pm 2.3$ |

