# OpenReview forum: "Functional Regularisation for  Continual Learning with Gaussian Processes"
_ICLR.cc/2020/Conference — Accept (Poster)_

### Official Review · AnonReviewer2 · 2019-10-22
**Official Blind Review #2**

**Rating:** 3

**Review:**

The paper develops a continual learning method based on Gaussian Processes (GPs) applied in the way introduced by prior work as Deep Kernel Learning (DKL). The proposed method summarizes tasks as sparse GPs and use them as regularizers for the subsequent tasks in order to avoid catastrophic forgetting. Salleviating the instability resulting from the representation drift.

Employing inducing point training for task memorization is a novel and interesting idea, which could be useful for the continual learning community. The fact that this approach also captures the uncertainty of the replays contributes fairly to robustness. Lastly, performing knowledge transfer by inheriting the KL term of the ELBO is also interesting, however, its theoretical implications deserve a close look. It would be enlightening to analyze which true posterior the learned model then corresponds to. Would not it be a slightly more principled Bayesian approach (i.e. one that has stronger grounds at first principles) to perform the knowledge transfer to assign the posterior of one task as the prior of the other, alternatively to keeping the entire KL term intact which employs the q(u_i) as the surrogate for q(u_j), i.e. the way introduced by Nguyen et al., 2017?

The presentation clarity of the paper is open for improvement. For instance, the abstract is written in a sort of convoluted way. I do not get how the KL divergence suddenly kicks in and for what exact purpose. Is it variational inference or a hand-designed regularization term?

I find the argumentation from Eq. 1 downwards until the end of Sec 2.1 on BNNs with stochastic weights and their relation to GPs a bit unnecessary complication. These are very well known facts. It would suffice to state briefly that the task learner is a vanilla DKL used within a vanilla sparse GP.

Figure 1 is also not so descriptive. I do not get what the GP here is exactly doing. What is input to and for which output modelity does it find a mapping? What is the calligraphic L in the figure? Is it a neural net loss or an ELBO?

In general I could not grasp why it makes sense to treat the the output layer params of a neural net treated for continual learning? They will not be sufficient to encode a task anyway, as an expressive enough neural net will leave only a linear mapping to the final layer. What happens if the intermediate representations of the input observations require a domain shift as the tasks evolve?

Overall, the presented ideas are fairly interesting and the experimental results are good enough for proof-of-concept, though not groundbreaking (behind the close relative VCL on MNIST and no comparison against VCL on Omniglot). Hence, this is a decent piece of work that lies somewhere around the borderline. My major concern is that the proposed method is conceptually not novel enough compared to Nguyen et al., 2017. My secondary concern is that the presentation is very much open to improvement in points hinted above.

--
Post-rebuttal: Thanks to authors for their tremendous effort to alleviate my concerns. The fact is that the conceptual novelty of the paper is too slim compared to VCL. As mentioned above, I even find the VCL approach more principled. I could have viewed the outcome of the paper as a slightly bigger news for the community if there was something unexpectedly positive on the reported results. However, as it appears from the comment below, the authors propose a super close variant of VCL that combines a few well-known techniques in a rather straightforward way and achieves in the end a model that performs on par with it. Under these conditions, I have hard time to find a reason to champion this paper for acceptance. That being said, I view this paper a tight borderline case due to its technical depth, hence I will not object to a reject decision either.

**Experience Assessment:**

I have published one or two papers in this area.

**Review Assessment: Checking Correctness Of Derivations And Theory:**

I assessed the sensibility of the derivations and theory.

**Review Assessment: Checking Correctness Of Experiments:**

I assessed the sensibility of the experiments.

**Review Assessment: Thoroughness In Paper Reading:**

I read the paper at least twice and used my best judgement in assessing the paper.

---

> ### Author Response · Authors · 2019-11-08
> **Response to Reviewer 2 (Part 1)**
>
> Thank you for the useful comments. We acknowledge that we could improve the manuscript to provide a more intuitive introduction and allow the reader to contrast weight-space and function-space regularisation. We would be grateful to hear if the reviewer has any concrete suggestions under which they would consider an increase in their rating.
>
> - Which true posterior does the learned model correspond to? Would not it be a slightly more principled Bayesian approach.... knowledge transfer to assign the posterior of one task as the prior of the other?
>
> The model learns a posterior distribution over each task-specific GP function f_i(x); Section 2.5 discusses the difference with weight-space posteriors. Regarding the second point, in Nguyen et al., 2017 it is important to distinguish between task-specific and shared parameters across tasks.  From a Bayesian perspective it is not correct for the posterior over task-specific parameters to act as prior over the task-specific parameters of the next task (and this is not done in VCL; see discussion on page 4 before Section 4 in https://arxiv.org/pdf/1710.10628.pdf ). In contrast, we fully agree with the reviewer that the posterior distribution over shared parameters indeed must be the prior for the next task.  In our formulation the only shared parameter is the feature vector parameter $\theta$ which is constantly updated by point estimation (not Bayesian inference), i.e the initial value $\theta$ for the next  task is the final value from the previous task and etc.  Therefore,  the comment  “transfer to assign the posterior of one task as the prior of the other” is consistent with the way we learn $\theta$, but instead of Bayesian inference we do point estimation for that parameter.  In contrast, all output weights $w_i$ and their corresponding function vectors $f_i$ (and the subset $u_i$), obtained by repametrising from the weight-space to the GP space, are task-specific parameters.  Therefore the variational posterior over $q(u_i)$ needs to be updated when we see data from the i-th task and it should not be the prior for the next task $i+1$.  Therefore, our method is principled from Bayesian perspective.
>
> To add more intuition, one can preserve our work as a mechanism for compressing data of previously seen tasks to the most significant data points that describe them. This is done individually (and sequentially) on each encountered task.  These compressed sets (inducing points) and their posterior distributions $q(u_i)$ are then used for replay (through each KL term $KL(q(u_i) || p_{\theta}(u_i))$)  in order to not forget the previously seen tasks. From this perspective it becomes clear that concatenating the different compressed sets (the inducing points) of each task is the right thing to do to represent all tasks, and that the last set of inducing points will only be useful to recover that particular task. We will try to make this explanation clearer in the main text of the paper.
>
> - Is it variational inference or a hand-designed regularization term?
>
> It is a principled variational inference procedure.  As in any variational Bayesian procedure in the ELBO  you automatically get KL divergence terms between posterior distributions and prior distributions (hence such terms also appears in VCL). Precisely the same holds for our method where each term $KL(q(u_i) || p_{\theta}(u_i))$ is the KL divergence between the posterior $q(u_i)$ and the prior $p_{\theta}(u_i)$. Each KL term regularises the shared representation parameters $\theta$ of the neural network, and therefore it regularises the full deep network.  The main difference however is that in our GP-based formulation the parameters correspond directly to values of the output function (each vector $f_i$ and its subset $u_i$ are values of the task-specific function $f_i(x)$) which leads to functional regularisation, i.e. regularisation by preserving posterior knowledge about  the task in the output function space, as opposed to weight space.  Therefore, our method is fundamentally different from methods like VCL and EWC that regularise based on the weight space (e.g. by using variational posteriors over task-specific weights $q(w_i)$). We have mathematically analysed this difference in Section 2.5; see also the discussion about the “moving target”  point 1) made by Reviewer 3 above.
>
> Description of Figure 1:
>
> We will improve the captioning of Figure 1, which is a very high level description. The calligraphic L is the loss for the corresponding task (e.g. negative log likelihood for classification, or L2 for regression). The diagram also makes explicit the features $\phi(x)$, which are simply features from a neural network. Thereafter, the output layer of our model is replaced by a GP. Once the task is learned, block B depicts what is means to define inducing points. Given these inducing points we can learn task 2, but now there is a regularisation term associated with task 1 (as explained above).

---

> ### Author Response · Authors · 2019-11-08
> **Response to Reviewer 2  (Part 2)**
>
>
> - Why consider only the output layer as a GP for CL:
>
> It’s important to note that the entire network is implicitly regularised by the loss function, even though the GP is only formulated for the last layer. Indeed, we are not posing any constraints about how individual weights can move during optimisation, as long as the network as a whole succeeds at explaining the current tasks and all functional regularisation terms. This is the fundamental difference between VCL or EWC (Kirckpatrick et. al, 2017 [https://arxiv.org/abs/1612.00796 ]) which instead explicitly regularise each parameter.
>
> This is actually interesting in the mentioned case where intermediate representations require a domain shift. Note that this will be necessary every time a new task is added, as the representations so far will not necessarily constitute optimal features for the next task. By allowing weights to vary freely as long as they explain all previous functions, we argue that intermediate representations can change more gracefully as opposed to the case where we force parameters to stay close to specific values.
>
> The entire network is regularized. We will make this more explicit in the main text. Why the mechanism of selecting the induced points is based on which datapoints are most significant to the GP (which encodes just the output layer), these data-points are then used to regularize the entire model.
>
> - Comparison to VCL on Omniglot:
>
> We agree that a comparison against VCL on Omniglot would be interesting. The primary reason this hasn’t been done in the initial submission is that reliable variational inference methods for CNNs (which are usually used for Omniglot) are yet to be developed. VCL relies on Mean-field VI which tends to not work very well for CNNs (see discussion here: https://openreview.net/forum?id=BkQqq0gRb ). We thus felt this was not a fair comparison, even though our method works well with both types of networks.
>
> Nevertheless, we are happy to conduct an ablation experiment on Omniglot for both VCL variants and our method. Due to the fairness of comparison concerns and the fact that the official VCL implementation has no code for CNNs, we will focus on MLPs for this comparison. We will report those results asap.

---

> > ### Author Response · Authors · 2019-11-13
> > **Response to Reviewer 2 (Additional experiments)**
> >
> > As promised, here are additional experimental results on Omniglot using an MLP. All experiments for VCL were run using the official implementation provided by the authors (https://github.com/nvcuong/variational-continual-learning/):
> >
> > ————————————————————————————————————
> > Algorithm                       |            Accuracy over all tasks at the end of training
> > ————————————————————————————————————
> > VCL (No coreset)           |	                         48.4 +- 0.7
> > ————————————————————————————————————
> >                                          |1 points/class | 2 points/class | 3 points/class
> > ————————————————————————————————————
> > VCL (Random coreset) |   49.18 +- 2.1   |	 50.50 +- 1.2	|   51.64 +- 1.0
> > VCL (K-center coreset) |   48.89 +- 1.1   |	 49.58 +- 1.4	|   49.61 +- 1.0
> > ————————————————————————————————————
> > FRCL (Trace)                   |   48.84 +- 1.1   |  52.10 +- 1.2	|   53.86 +- 2.3
> > ————————————————————————————————————
> >
> > For all experiments, we used an MLP with 4 hidden layers of 256 units each and ReLU activations, a batch size of 100 and the Adam Optimiser (Step size of 0.001 for VCL and 0.0001 for FRCL). We optimised  both types of algorithms independently and found the following parameters:
> >
> > VCL: 100 training epochs per task, 50 adaptation epochs to coreset*, Multi-Head
> > FRCL: 1500 training steps per task, 2000 discrete optimization steps, inducing points initialised as a uniform distribution over classes.
> >
> > * VCL with coresets relies on training task-specific (in the multi-head case) and shared parameters on the coreset of each task before evaluation. This means that for Omniglot, the algorithm eventually requires 50 copies of the same network.
> >
> > Does this answer the reviewer's questions?

---

### Official Review · AnonReviewer3 · 2019-10-22
**Official Blind Review #3**

**Rating:** 6

**Review:**

The authors propose a function-space based approach to continual learning problems (CL), wherein a learned embedding

    $\hat{\mathbf{x}} = \text{NN}(\mathbf{x}; \theta)$

is shared between task-specific GPs s.t.

    $f_{i}(\mathbf{X}) \sim \mathcal{N}(\mathbf{0}, k_{i}(\hat{\mathbf{X}},\hat{\mathbf{X}}))$,

where the $i$-th task's covariance $k_{i}$ is a defined via standard variational inducing points methods. CL manifests as KL divergences between tasks' variational posteriors $q_{i}$ and their respective priors $p_{i}$. Since the embedding helps define $p_{i}$, its parameters $\theta$ are regularized to promote sharing.

The work investigates both practical and theoretical implications of this setup. On the practical side, the authors discuss enhanced 'on-task' inference via hybridization of function- and weight-space based approaches and, subsequently, strategies for optimizing inducing points. Additionally, a novel approach for automatically detect task switching is introduced that exploits the Bayesian aspects of the proposed framework.

On the theoretical side, points of (personal) interest revolved around differences between weight- and function-space approaches to CL. Here, I think that streamlining the presented argument would go a long ways. Paraphrasing, one of the authors' key insights is that:

  1) CL in weight-spaces is hard, since weights' semantics are moving target that change along with shared parameters.
  2) CL in function-space is easy, since the functions (i.e. tasks) themselves remain the same.

This information is provided in the introduction, but (as a relative newcomer to CL) I failed to connect regularization and rehearsal/replay based methods with the aforementioned spaces. It was only upon reading Sect 2.5 that this intuition 'clicked' for me. Hence, I suggest making this observation as obvious and intuitive as possible.

The provided experiments seem reasonable and do a good job highlighting different facets of the paper. Two additional results would be appreciated:

  a) How well calibrated are FRCL-based classifiers?
  b) How impactful is the hybrid representation (Sect 2.3) for test-time performance?

GP approximations formulated solely in terms of weighted sums of (finitely many) basis functions typically suffer from degradation of predictive uncertainties. Since one often motivates use of GPs via a desire for well-calibrated uncertainty, (a) seem quite pertinent.


Nitpicks, Spelling, & Grammar:
  - Lots of run-on sentences; consider breaking these up.
  - Introductory modifying phrases are missing commas.
  - Consider citing other recent works that use NN basis functions in conjunction with Bayesian Linear Regression.
  - Various missing or superfluous words resulting in some garbled sentences, e.g.:
      - "... our approach looks constraining."
      - "The ability to detect changes based on the above procedure comes from that in"
      - "While the task boundary detection results for Omniglot are less strong, which may due to the smaller batch size (32 for Omniglot, ≥ 100 for the MNIST-versions), resulting a noisier test result."


**Experience Assessment:**

I have read many papers in this area.

**Review Assessment: Checking Correctness Of Derivations And Theory:**

I assessed the sensibility of the derivations and theory.

**Review Assessment: Checking Correctness Of Experiments:**

I assessed the sensibility of the experiments.

**Review Assessment: Thoroughness In Paper Reading:**

I read the paper at least twice and used my best judgement in assessing the paper.

---

> ### Author Response · Authors · 2019-11-08
> **Response to Reviewer 3**
>
> Thank you for your thoughtful review . Point 1) and 2) describes well the differences between regularising continual learning in the function space rather than on the weight space. To add to that, the motivation behind our method is that  learning a supervised task corresponds to learning a function, and thus our method tries to “remember” the task by remembering direct posterior estimates over output values of that function at informative inputs.  The  “moving target” comment made by the reviewer regarding methods that regularise based on posteriors over weights captures precisely the intuition of what mathematically we analyse in Section 2.5 and we are glad that the reviewer found this useful. Given that Section 2.5 provides a more technical explanation, we will try to follow reviewer suggestions and provide a more intuitive discussion earlier in the paper.
>
>
> - Robustness on FRCL-based classifiers and impactfulness of hybrid representation for test-time performance:
>
> It is hard to know how well calibrated are the uncertainties of FRCL since we do not know the ground-truth. FRCL is based on a neural network where we do Bayesian inference only over the final layer weights (Usually termed Deep Kernel learning (Wilson et al., AISTATS 2016) [http://proceedings.mlr.press/v51/wilson16.pdf ]). It is encouraging that according to the large scale study in (Riquelme et. al, 2018 [https://arxiv.org/abs/1802.09127 ]) such as (not fully Bayesian approach) is always one of the best techniques among many other methods in contextual bandits applications, where modelling well uncertainties within a Thompson sampling exploration framework is very crucial. The aforementioned work by (Wilson et al., AISTATS 2016) also shows that predictive uncertainty can be of high quality. Regarding b) if we do not apply the hybrid approach where inference over the current task is done in the weight-space, and instead we apply variational sparse GPs for the current task, the performance is significantly worse (we can add a Table for that in the Appendix for completeness). The reason is that the estimate of the posterior distribution $q(u_i)$ over the inducing values is not accurate enough both in terms of the mean and also in terms of variances (typically underestimation) when compared to $q(u_i)$ obtained by the hybrid approach. The latest  allows fitting the current task with the tightest possible ELBO and getting the best possible approximate posterior  (i.e. with no additional approximation error due to the inducing points and the variational sparse GP) which leads to better estimate of each $q(u_i)$ and subsequently better regularisation for continual learning.
>
> - GP approximations in terms of finitely many basis functions:
>
> The degradation of predictive uncertainties of finite basis functions certainly occurs when the basis function are local, e.g radial basis functions, but typically does not occur for non local basis functions/activation units  as the ones we typically use in neural networks. E.g. when the feature vector  is defined by ReLUs or tanh activation functions, which are non local, the degradation of predictive variances is typically not observed as we move away from the training inputs. This is to some extent also confirmed by the task-boundary detection results.

---

### Official Review · AnonReviewer1 · 2019-11-04
**Official Blind Review #1**

**Rating:** 6

**Review:**

Summary:
The authors propose a method to perform continual learning with neural networks by incorporating variational Gaussian Processes as a top layer (also called Deep Kernel Learning) and constructing an objective utilizing the inducing inputs and outputs to memorize across tasks.
They further study ways to approximate this behavior with weight space models and use their model for task boundary detection by utilizing statistical tests and Bayesian model selection.
Experiments show good performance of their method.

Comments:
1. The mathematical formulation of the basic model is very elegant. However, it is not immediately clear to me that the joint ELBO across successive tasks is still lower-bounding the actual objective.
2. The paper is well written overall.
3. To the best of my knowledge using such a model for task boundary detection is novel and quite interesting. There are obvious links to Bayesian changepoint detection in the timeseries setting. Possibly these links would be made more clear by a citation to a recent paper such as Spatio-temporal Bayesian On-line Changepoint Detection with Model Selection by Knoblauch and Damoulas, or any other paper of similar content. The link is quite fascinating.
4. Sections 2.3 and 2.4 of the paper are the weakest points and quite unsatisfactory as they forgo the elegance of the proposed approach to do "something else" that Sec. D explains how to salvage with "tricks". Especially with regards to Sec. 2.4, why can't we just do inference on Z_i and have to pick datapoints via discrete optimization? That comparison would be useful in the experiments. Furthermore, recent papers utilizing GPytorch by Gardner et al have dramatically sped up GP inference. Could we aim to make the original idea fast enough to be used instead of resorting to an approximate model with weight spaces and corrections to extract Z and u per task?
5. The experiments are good, but very focused on MNIST tasks. I would appreciate tasks of different structure given how well the method appears to work.


Decision:
I find the basic idea of the paper quite appealing as it leverages the elegance of the deep kernel learning formulation to yield an attempt at a principled Bayesian version of continual learning and demonstrates empirical value.
Some discussion on the objective might be warranted to demonstrate that it actually lower bounds the true LLK.
I am quite happy with the task boundary detection section and would encourage the authors to strengthen the link to changepoint detection.
My biggest qualms with the paper are that it departs from that strategy and performs weight space inference for training per task and then "corrects" to move back to the GP representation. A more convincing discussion would be welcome here.
The experiments are functional and show good results, but I would appreciate more diversity in the tasks.
As the paper stands I learn towards recommending acceptance and would strongly encourage the authors to iron out the weaknesses of the paper.

**Experience Assessment:**

I have published in this field for several years.

**Review Assessment: Checking Correctness Of Derivations And Theory:**

I carefully checked the derivations and theory.

**Review Assessment: Checking Correctness Of Experiments:**

I assessed the sensibility of the experiments.

**Review Assessment: Thoroughness In Paper Reading:**

I read the paper thoroughly.

---

> ### Author Response · Authors · 2019-11-08
> **Response to Reviewer 1**
>
> Thank you for your excellent comments. We address questions below:
>
> - Is the joint ELBO across successive tasks still lower-bounding the actual objective?
>
> As mentioned in Section 2.2 the ELBO across successive tasks is only an approximation to the full ELBO obtained under the constraint that we only maintain a subset  $Z_1$ and not the  full training set for the task 1. When $Z_1$ (and similarly each $Z_k$ for more than two tasks) is a random subset of $X_1$ then the approximation is, up to a constant, an unbiased approximation to the full exact ELBO (constructed similarly to stochastic/minibatch variational inference; see Hoffman et al. 2013 [http://www.columbia.edu/~jwp2128/Papers/HoffmanBleiWangPaisley2013.pdf]).  This means that such an approximation might not be a strict lower bound on the exact log marginal likelihood (over the full data that are not kept in memory), but it lower bounds this log marginal likelihood stochastically, i.e. on expectation under the distribution associated with  selecting a random $Z_k$ from $X_k$.   When $Z_k$ is not chosen randomly we might lose the unbiasedness property, but we might get better approximation for performing continual learning since intuitively we want to maintain the most informative $Z_k$, as inducing inputs, for each task in order to avoid forgetting.
>
> - Changepoint detection:
>
> Thanks for the reference. We agree that the connection to Changepoint detection is very interesting and will make this more precise. Note that the works by e.g. (Adams and Mackay (2007)[https://arxiv.org/abs/0710.3742] and Fearnhead (2006) [https://eprints.lancs.ac.uk/id/eprint/8189/]) are actually straight-forwardly applicable to the time series shown in (Fig. 5), which would be an elegant alternative to the t-test used in our initial submission. We will strive to include those results)
>
> - Regarding Section 2.3 and 2.4:
>
> Thank you for this comment. Regarding the weight space inference for the current task that we do in Section 2.3 this step is rigorous since we never change or approximate the exact GP model, we simply reparametrize it.  This is because the special form of the linear kernel $k(x,x’) = \phi(x;\theta)^\top \phi(x’;\theta)$ allows to express two equivalent representations of the model:  (i) over the weights $w_i$ and (ii) over the full vector of training function values $f_i$.  The exact marginal likelihood can be written as
> $$
> p(y) = \int p(y|w_i) p(w_i) d w_i =  \int p(y_i|f_i) p(f_i) d f_i,
> $$
> where the second integral (the standard form for the exact GP marginal likelihood) is obtained by reparametrizing $f_{i,j} = w_i^\top \phi(x_j;\theta), j=1, \ldots,N_i$.  This proves that an ELBO based on the first integral (over $w_i$) is an ELBO on the exact GP marginal likelihood.  However, there are important computational differences. If you work with $f_i$ the complexity per optimisation step is $O(N^3$) but if you work with $w_i$  complexity is $O(b K^2)$, where $b$ is the minibatch size and $K$ the size of the feature vector.  Both approaches lower bound the same exact GP marginal likelihood and therefore they approximate the same exact GP model (so the statement “resorting to an approximate model with weight spaces” is not true since inference over $w_i$ corresponds to inference under the exact model).  The theoretical  equivalence  and computational differences between inference in function space and weight space for this particular type of models is discussed e.g. in (Williams, 1998)  http://citeseerx.ist.psu.edu/viewdoc/download?doi=10.1.1.84.1226&rep=rep1&type=pdf and in the Rasmussen and Williams book.
>
> Regarding Section 2.4 we have chosen to perform discrete optimisation over the inducing points because we believe that continuous optimisation over the inducing inputs will be hard in high dimensions. E.g consider Figure 4 where the panel shows the initial inputs and panel c the final ones, found under discrete optimisation. While continuous gradient-based optimisation over the inducing inputs is in general very difficult, it may in practice be possible to initialise the continous optimisation procedure by first taking a few steps of discrete optimisation.

---

### Decision · Program_Chairs · 2019-12-19

**Decision:**

Accept (Poster)

**Comment:**

The authors introduce a framework for continual learning in neural networks based on sparse Gaussian process methods. The reviewers had a number of questions and concerns, that were adequately addressed during the discussion phase. This is an interesting addition to the continual learning literature. Please be sure to update the paper based on the discussion.